# Diagnostic Performance of Selected MRI-Derived Radiomics Able to Discriminate Progression-Free and Overall Survival in Patients with Midline Glioma and the H3F3AK27M Mutation

**DOI:** 10.3390/diagnostics13050849

**Published:** 2023-02-23

**Authors:** Maria-Fatima Chilaca-Rosas, Melissa Garcia-Lezama, Sergio Moreno-Jimenez, Ernesto Roldan-Valadez

**Affiliations:** 1Radiotherapy Department, Hospital de Oncología, Centro Medico Nacional Siglo XXI, Instituto Mexicano del Seguro Social, Mexico City 06720, Mexico; 2Directorate of Research, Hospital General de Mexico “Dr Eduardo Liceaga”, Mexico City 06720, Mexico; 3Directorate of Surgery, Instituto Nacional de Neurología y Neurocirugia, “Manuel Velasco Suarez”, Mexico City 14269, Mexico; 4Department of Radiology, I.M. Sechenov First Moscow State Medical University (Sechenov University), Moscow 119992, Russia

**Keywords:** midline glioma, paediatric, brain tumour, radiomic, MRI, prognosis

## Abstract

Background: Radiomics refers to a recent area of knowledge that studies features extracted from different imaging techniques and subsequently transformed into high-dimensional data that can be associated with biological events. Diffuse midline gliomas (DMG) are one of the most devastating types of cancer, with a median survival of approximately 11 months after diagnosis and 4–5 months after radiological and clinical progression. Methods: A retrospective study. From a database of 91 patients with DMG, only 12 had the H3.3K27M mutation and brain MRI DICOM files available. Radiomic features were extracted from MRI T1 and T2 sequences using LIFEx software. Statistical analysis included normal distribution tests and the Mann–Whitney U test, ROC analysis, and calculation of cut-off values. Results: A total of 5760 radiomic values were included in the analyses. AUROC demonstrated 13 radiomics with statistical significance for progression-free survival (PFS) and overall survival (OS). Diagnostic performance tests showed nine radiomics with specificity for PFS above 90% and one with a sensitivity of 97.2%. For OS, 3 out of 4 radiomics demonstrated between 80 and 90% sensitivity. Conclusions: Several radiomic features demonstrated statistical significance and have the potential to further aid DMG diagnostic assessment non-invasively. The most significant radiomics were first- and second-order features with GLCM texture profile, GLZLM_GLNU, and NGLDM_Contrast.

## 1. Introduction

Diffuse midline gliomas (DMG) are aggressive astrocytic tumours that occur in the paediatric population, adolescents, and young adults (AYA) affected in the brainstem or midline structures [1,2]. It has a poor median survival of approximately 11 months from histopathological diagnosis and only 4–5 months after radiological and clinical progression [3,4].

Advances in molecular diagnosis have had an essential impact in refining the discrimination of tumours according to genetic characteristics. We can particularly distinguish those DMG with the H3.K27M mutation since it is associated with a worse outcome, showing a survival rate after two years lower than 10% [5].

“Radiomics” refers to the acquisition of traces of quantitative features that are usually non-perceptible to human vision and are obtained from different imaging techniques that are subsequently transformed into high dimensional data that can be associated with biological events such as the progression of tumours in oncology [6]. Radiomics encompasses novel methods for high-throughput quantitative feature extraction from a pixel value, distribution, spatial relation, or specified region of interest (ROI) and volume of interest (VOI) from conventional and advanced imaging modalities [5], and in recent years it has been of interest in brain tumours such as gliomas.

The past decades have had a considerable impact in the medical imaging field; previously perceived qualitative-only images can now be minable quantitative data rather than pictures [7]. Furthermore, recently they have been found to predict histology, response to treatment, genetic signature, recurrence, and survival, among other features, in several pathologies, especially cancer [8,9]. In a recent meta-analysis, radiomics depicted a high level of predictive value for pre-operative lymph node metastasis in patients with biliary tract cancers; this study also showed that MRI-derived radiomics had a higher pooled sensitivity than CT [10].

Radiomics can also potentially be used for personalised therapy and the prediction of individual treatment outcomes [7,11]. Moreover, with the increasing implementation of artificial intelligence (AI) in medical imaging, this could go even further; however, this will not be further discussed as it is out of the scope of our research and implicates psychological, ethical, and medico-legal issues that need to be addressed for AI to be applicable in patient management [12].

Although radiomics demonstrated potential in survival prediction for DMG from presurgical neuroimaging using a connectome model in 2022 [13], this study did not show which specific radiomic could support the predictive ability. Overall, there is still scarce research on the subject. Until February 2023, there were only nine publications on PubMed, including “Radiomics” and “Midline gliomas or midline glioma” in their title; none of them presented a diagnostic performance test with cut-off values for specific radiomics [5,14,15,16,17,18,19,20,21].

A study published by Wagner et al. in 2023 evaluated MRI-derived radiomic features able to predict PFS and OS. It included a cohort of 89 patients; however, in their study not all patients had the histopathological and molecular diagnosis (29.2%), which, as mentioned by themselves, is inherent to most DMGs. They identified radiomics of T2-FLAIR and nonenhanced T1-weighted sequences that were predictive of PFS and GLSZ as the predominant predictive radiomic feature matrix. It was also mentioned that incorporating H3 K27M molecular markers could further increase model performance [22].

Our goal with this study was to perform a comprehensive diagnostic assessment of MRI-derived radiomics in patients diagnosed with DMG using conventional MRI sequences (T1 post gadolinium and T2) to calculate the basic diagnostic performance parameters from the set of radiomics of one of the leading oncology semi-automatised software programs [23]. Select parameters might support the assembling of specific predictive modelling using radiomic-specific cut-off values.

## 2. Materials and Methods

This observational retrospective HIPAA-compliant study was approved by the National Committee of Scientific Research of Oncology and paediatrics hospital of National Medical Centre S.XXI IMSS, Mexico City with No. F-CNIC-2020-321; it included patients’ records with midline gliomas identified in paediatrics and AYA in the database of Research of Oncology and paediatrics hospital of National Medical Centre S.XXI IMSS, Mexico City. All the information reported followed the national and international standards for the management of clinical files (the official Mexican norm NOM-012-SSA3-201218, which establishes the criteria for the execution of scientific projects for the health of human beings) and with the Helsinki Declaration of 1975, as revised in 2013.

Inclusion criteria included a histopathological diagnosis of midline glioma with the determination of the H3.3K27M mutation according to WHO 2007 and 2016 and the tumour’s location in the midline region based on magnetic resonance imaging (MRI) [12]. The exclusion criteria were concurrence with other neoplasms from 5 years ago or another lethal comorbidity. Elimination criteria were applied to MRI studies whose DICOM files could not be downloaded for assessment. Of the initial 91 patients identified, 46 underwent a biopsy procedure but 26 presented undetermined results due to insufficient tissue samples. Two tissue samples were negative from the remaining 20 patients, and 18 were positive for the HF3K27M mutation, fulfilling the 2007 and 2016 WHO criteria for the histopathological diagnosis. Of the 18 positive patients, only 12 had available DICOM MRI files included in the analysis. The other six patients with positive HF3K27M mutation had only DICOM files of computed tomography and, therefore, were eliminated from the study. A flow diagram of the patient selection is shown in Figure 1.

### 2.1. Imaging Evaluation of Midline Gliomas

MRI evaluations were conducted on a 1.5T Magnetom (Siemens Healthcare, Erlangen, Germany) and a 3T Magnetom (Philips Health, Hong Kong, China). Gadolinium-enhanced T1-weighted images (T1WI) covering the entire midline and T2-weighted images were acquired with 60–80-slice processing. The MRI evaluation was performed by two neuroradiologists and a radio-oncologist with over ten years of experience. The radiological features involved tumour location, peritumoral oedema and necrosis, tumour volume, and cystic changes. The T1 pre- and post-gadolinium and T2 sequences allowed the identification of morphological features (see Figure 2). The images were processed for analysis in the semiautomatic platform, and the radiomic characteristics were obtained, which were later analysed with the oncological results (see Figure 3).

### 2.2. Radiomics Extraction

We used the software LIFEx v7.1.0 (French Alternative Energies and Atomic Energy Commission, Orsay, France) to extract radiomic features from the brain MRI images of patients with DMG. It is a software for radiomics feature extraction in multimodality imaging. Until the writing of this article (27 January 2023), the official website for the software reported its use in 497 publications in PubMed; it also received 633 citations with special recognition in oncology [24]. The methods for using the LIFEx v7.1.0 have been previously reported and contemplate the IBSI criteria [23]. MRI images of the T1 pre- and post-gadolinium and T2 sequences were uploaded into the LIFEx to extract the radiomic characteristics; it extracted 94 different radiomics, which were grouped into three categories: shape and volume features, intensity features, and texture features. From the total of 94 radiomics extracted, 14 were eliminated from our analyses since they apply to other imaging techniques (CT, NM, and PET), leaving a total of only 80 radiomics that were included in our calculations. In each of the selected brain MRIs, three regions were selected for extracting the radiomics: viable tumour (represented for the enhancing tumour regions in T1 post-gadolinium sequence), peritumoral oedema (visualised on T2), and equivalent midline normal tissue (EMNT) observed in both sequences. The ROI and VOI were manually drawn across all slices of the selected sequences and validated by the imaging experts.

### 2.3. Definitions of Progression-Free Survival and Overall Survival

PFS definition was established as the time elapsed from completing radiotherapy until tumour progression or death. The pre-operative MRI was the base for radiologists and radio-oncologists, which aided in monitoring disease evolution. The tumour recurrence report was based on the assessment of T1 post-gadolinium MRI, describing new or increased enhancing tumours within the site of initial surgical resection. RANO criteria were also considered to discriminate actual progression or recurrence against pseudoprogression when repeat pathology was unavailable. Progressive disease was defined as a ≥25% augmentation in enhancing tumour along with clinical deterioration requiring treatment change within six months post-RT [25]. Overall survival was defined as the time elapsed from the MRI diagnostic to the time of death from any cause [26].

### 2.4. Statistical Analyses

#### 2.4.1. Descriptive Statistics and Differences between Tumour Regions

The value distribution of each radiomic was analysed using the Kolmogorov–Smirnov and Shapiro–Wilk tests. Because most of the values depicted a non-normal distribution, we used quartiles and the interquartile range (IQR) to represent the central tendency and dispersion of data. To explore simultaneous differences among regions (tumour, peritumoral oedema, and EMNT), we used the Kruskal–Wallis test followed by the Mann–Whitney U test. Significant statistical differences for ROC curve analyses were demonstrated by a *p*-value < 0.05.

#### 2.4.2. Evaluation of Diagnostic Performance

Following previous publications about the diagnostic performance of quantitative brain biomarkers [27,28], the area under the ROC curve (AUROC) was considered the measurement for the overall performance for each selected GM or WM region, or GM/WM ratios [29]. Additionally, the standard error (SE), *p*-value, and 95% confidence interval (CI) were described. For the calculation of SE, the method of Delong [30] was applied. Other important values displayed are the sensitivity, specificity, positive and negative predictive values, and positive and negative likelihoods. Cut-off ranges served as the ideal cut-off to assess the presence of a clinical diagnosis by maximising sensitivity and specificity. To determine the accuracy, a traditionally established academic points system classification was applied as follows: 90–100% = excellent, 80–90% = good, 70–80% = fair, 60–70% = poor, and 50–60% = fail [31]. Data was presented following the Standards for Reporting of Diagnostic Accuracy (STARD) initiative [16].

#### 2.4.3. Software

The analyses in this study were performed using IBM SPSS^®^ (Version 27.0.0.1, International Business Machines Corp., Armonk, NY, USA) and MedCalc^®^ (Version 20.0.15 MedCalc Software bvba, Mariakerke, Belgium).

## 3. Results

### 3.1. Demographics and Clinical Features

Of the 12 patients included, 9 were paediatric (75%) and 3 were AYAs (25%). The relation between male and female patients (M: F) was 2:1. Tumours were more frequently located in the pons (50%). The rest of the tumours were equally found in the thalamus (25%) and midbrain (25%). Patients underwent radiotherapy with a median dose of 55 Gy (ranging from 54 to 55 Gy). The radiotherapy schemes were conventional in 83.3% of patients and hypofractionated in the remaining 16.7%.

### 3.2. Survival Analysis

Kaplan–Meier curves showed that the median PFS was 210 days (7 months). They also depicted that the median OS was 455 days (15.2 months) in the group of patients in this study (see Figure 4).

### 3.3. Radiomic Values Differences between Tumour Regions

The Kruskal–Wallis H test was performed but did not depict any statistically significant differences between the tumour regions of each radiomic variable (*p* > 0.05).

### 3.4. Radiomic Values Extraction

From the three tumour regions of DMG (viable tumour, peritumoral oedema, and EMNT), 80 radiomics were obtained from each region. These measurements represented 240 radiomic features for the T1 post-gadolinium and 240 radiomics for the T2 sequence per patient. A total of 5760 radiomic values were included in the analyses. Table 1 shows the quartiles and IQR for the selected radiomics.

### 3.5. Radiomics with Significant AUROCs to Identify PFS and OS above the Median

Our analysis showed 13 radiomics with a statistically significant difference in AUROC (*p*-value < 0.05). See Table 2 part A for the list of significant radiomics discriminating PFS. We found ROC curves for four radiomics with a statistically significant difference in AUROC (*p*-value < 0.05). See Table 2 part B for the list of significant radiomics discriminating OS. Figure 5 shows the performed ROC curves.

### 3.6. Diagnostic Performance Test

According to the mentioned traditional academic points system, our assessment found that 9 out of the 13 significant radiomics showed excellent specificity for PFS. For sensitivity, one radiomic fell under the excellent (above 90%) category (CONVENTIONAL_peakSphere0.5mLdiscretizedvolumesought), and there was one in the good (80–90%) category (DISCRETIZED_peakSphere0.5mLdiscretizedvolumesought).

For OS, three out of four significant radiomics demonstrated good (80–90%) sensitivity (CONVENTIONAL_peakSphere 1mLdiscretizedvolumesought, CONVENTIONAL_RIM_stdev and DISCRETIZED_peakSphere 1mLdiscretizedvolumesought); only the radiomic DISCRETIZED_Q1 showed fair (70–80%) specificity on the performance test. A complete description of the diagnostic performance tests is presented in Table 3.

## 4. Discussion

DMG is one of the most devastating types of cancer, in which its intrinsic location contributes to a poor prognosis. Recent findings have entirely revolutionised the knowledge of this disease, such as the H3.K27M mutation in up to 70% of DMG samples. However, despite the new findings, clinical trials of promising therapies, and current management involving radiotherapy and chemotherapy, the overall oncological results are still limited as the 2-year survival rate remains <10% [1,3]. In the last decade, quantifications of multiple biological events have been sought in this new area of knowledge and, in the case of neuro-oncology, with a more significant challenge of establishing objectives such as: (A) differential diagnosis, as published by Suh H. et al. 2018, between tumours glial and central nervous system lymphomas; (B) types of grade focusing on glial astrocytic tumours, the differentiation of low-grade vs. high-grade tumours; (C) vascularity mapping; and (D) responses to treatments (progression vs. pseudo-progression) [32,33,34,35]. 

Although preliminary radiomics studies have shown promising results related to oncology results, such as survival [36], there is no consensus for its semi-automated or complete automated application. Most published radiomics studies present signatures (clinical results with radiomic models) in addition to using an independent test set. It is a semi-automated system with segmentation by neuroimaging, and neuro-oncology experts have proposed a tendency toward acceptance [37].

Wagner et al. presented a study of a conditional survival forest model applied to predict progression-free survival (PFS) using training data. Among their findings for PFS, the best FLAIR radiomics model yielded a concordance of 0.87 at four months, the best T1-weighted radiomics model yielded a concordance of 0.82 at four months, and the best combined FLAIR + T1-weighted radiomics model yielded a concordance of 0.74 at three months PFS [22]. However, molecular markers such as the H3K27M mutation were not incorporated due to the partial lack of biopsies in their cohort. This emphasises the importance of continuing this research line and further establishing non-invasive diagnostic/prognostic parameters that may aid clinicians.

In this study, we were able to complete the aims proposed at the beginning of this manuscript: to identify which radiomics extracted with the software LIFEx were able to discriminate PFS and OS in patients with DMG and the H3F3AK27M mutation and to obtain a comprehensive diagnostic performance assessment of those selected radiomics. The clinical relevance of this study consists of identifying cut-off values for 17 radiomics with statistical significance. From 70 radiomics initially evaluated, 13 radiomics were significant in determining patients with PFS, whereas four radiomics were significant in discriminating patients above the median of OS.

The radiomic characteristics were obtained in sequence: description characteristics, first-order statistics (histogram representation of ROI and VOI and calculation of mean, percentile, skewness entropy, and kurtosis), as well as second-order statistics (mainly assess patterns related to spatial distribution and co-occurrence of pixels and voxels) and later deriving to the radiomic characteristics of standard textures: grey-level co-occurrence matrix (GLCM) and grey-level run-length matrix (GLRM) [38]. With the analyses mentioned above, we can generate valuable information that is not perceived by human perception and that translates data from the tumour microenvironment to us [39,40], providing information that would have great predictive potential, such as the results and trends that we present in this paper [41].

In our research, the description characteristic related to the progression of cases such as PARAMS YSpatialResampling is associated with the voxel dimension in the images, for which it would be essential to resume the use of digital imaging terms in daily clinical practice with the medical images of the patients and the relevance of voxel. Further, four radiomics derived from GLCM were significant; contrast variance (of heterogeneity that gives more weight to pairs of different intensity levels that depart from the mean), correlation (shows the linear dependency of grey-level values to their corresponding voxels in the GLCM), entropy (a statistical measure of randomness), and joint entropy (a measure of the randomness/variability in neighbourhood intensity values) [42]. Regarding the second-order radiomic characteristics that have been evaluated with greater solidity in survival prognosis since the work of Molina et al. 2016, it has been with low levels of GLCM_ entropy and GLCM_ contrast, although in 3D analysis/manual of studies of patients with glioblastomas, which in our work we evaluated with the 3D semi-automated platform and observed significance related to progression-free survival of the radiomic characteristics GLCM_ContrastVariance and GLCM_correlation. A recent study published in January 2023 depicted the ability of GLCM radiomics to predict complete pathological responses in patients diagnosed with invasive breast cancer [43].

To emphasise the importance of the information obtained by radiomics, a need to implement the use of digital image terms such as voxels and discretised volumes, which are more precise than even cubic centimetres and more practical with values and cut-off points than support the best clinical practice for neurosurgery and radiotherapy management planning, was proposed. We observed a trend in this work suggested by other publications [44,45].

It has been described in the literature how radiomics models have the potential to assist clinicians in improving confidence in decision making in oncologic management [46]. Additionally, the publication of models depicting their verified negative and positive predictions of radiomics will have great potential to assist radio-oncologists, radiologists, and oncologists.

### 4.1. Limitations of the Study

Several limitations need to be addressed. We acknowledge the limited number of patients analysed. We found a lack of international consensus about the requirements and restrictions in extracting radiomic characteristics. The short evolution of the patients means that minimal information on the cases is described in clinical practice and, due to the neurological deterioration in the progression, fewer than expected studies can be obtained, including functional ones such as DTI, to evaluate more characteristics. In many cases, even performing an MRI is limited due to severe conditions, neurological deterioration, involuntary movements, and lack of time for the imaging assessment. Several patients identified were omitted because they had only undergone CT scans.

The scarcity of public databases prevents the analysis and comparison of data in search of the standardisation of strategies for the diagnostic and prognostic approach in these orphan pathologies, such as midline gliomas, in addition to the lack of a consensus for segmentation guides according to the published evidence of the ROI and radiomic profiles associated with prognosis and diagnosis.

Furthermore, in developing countries such as ours, accessibility to medical services affects early diagnoses and generates a concentration of patients with impaired initial functional status.

We did not use the radiotherapy targets gross tumour volume (GTV) or clinical target volume (CTV) contoured by the radio-oncologist manually, since these two variables were out of the scope of the research, as this could easily extend onto another manuscript.

It is important to remark that our study was performed with two MRI scanners. Our population could not perform the same analyses under more MRI scanners. Prior studies have evaluated the variability of results in radiomic studies according to the MRI and its selected scanner parameters [47,48,49,50,51]. The variations found are called the “scanner effect” and require harmonisation [52]. However, methods for bias correction to avoid this effect have also been studied with promising results [53,54,55]. We know this limitation needs to be better standardised for further application for radiomic models. 

A severe difficulty we encountered during this research was the lack of descriptive studies that specifically focused on some of the mentioned radiomics (PARAMS_YSpatialResampling, Conventional_peaksphere’s). This is the reason why we were unable to compare our findings with other studies for all the radiomic features. To our knowledge, most studies report findings according to radiomic subgroups but not their performance.

### 4.2. Future Directions

For a better understanding and more complex research on orphan entities, it would be ideal to have an international data concentration centre where researchers from institutions worldwide could include their radiomic measurements from studies with a small number of patients. In this study, we were surprised that from only 12 patients and three tumour regions, over 5700 radiomic measurements were extracted. The amount of data that a multicentre study would generate with over 100 patients will also require dedicated computational power and a consensus for the radiomic measurements, extraction, analyses, use of machine learning, and other AI techniques.

Future studies should focus on the discretisation of the volumes and the base textures that, in this pathology, suggests quantified subunits of heterogeneities that will be relevant for the planning of oncological management [40]. There is a need for multicentre worldwide collaboration to find clues for progress and a race to improve these results for this group of patients. Proposals for algorithms and decision trees as guides for daily clinical practice with the evidence that is generated with more significant support by current publications in a cautious manner, mainly in an orphan entity such as the current one. To establish a better consensus, further studies involving more extensive databases of patients are necessary, including patients with different variations of the H3F3A mutations.

## 5. Conclusions

There is a statistical trend for the role of first- and second-order radiomic features in influencing PFS and OS in this orphan entity. There is relevance for the GLCM texture profile, GLZLM_GLNU, and NGLDM_Contrast. The information found in this study could add to the existing evidence in the literature that would enrich the decision trees in the diagnosis and treatment of precision with the neuro-oncology team (neuroradiologists, neuropathologists, neurosurgeons, neuro-oncologists, paediatricians, and neurologists) and better support and perform precision medicine with fewer invasive procedures and the dose-paint technique for future radiotherapy treatment planning.

## Figures and Tables

**Figure 1 diagnostics-13-00849-f001:**
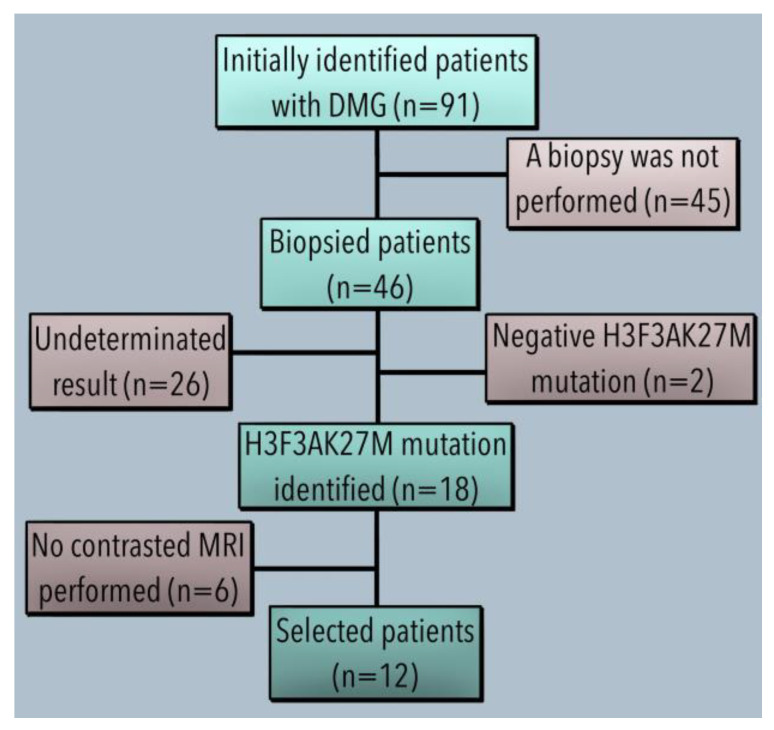
Patient selection process.

**Figure 2 diagnostics-13-00849-f002:**
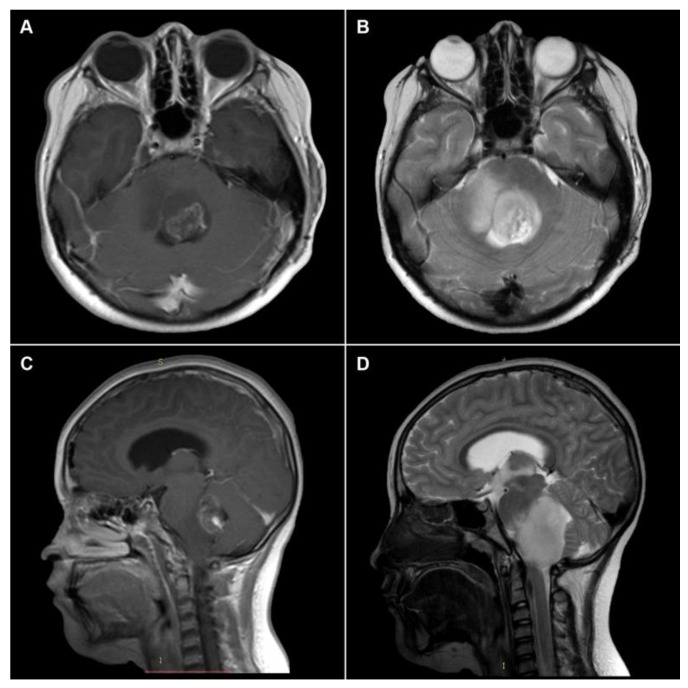
The axial and sagittal planes show the MRI appearance of DMG on T1- and T2-weighted images. (**A**,**C**) Images show contrast-enhanced tumoural lesions, but (**B**,**D**) images showed extensive oedema tumoural and diffuse/infiltrative patterns.

**Figure 3 diagnostics-13-00849-f003:**
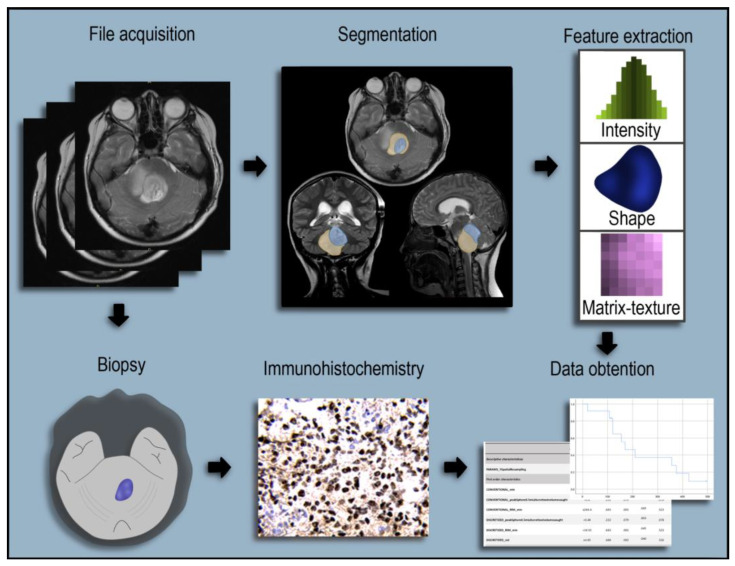
Description of steps followed for the data acquisition of patients. In the initial stage, patients underwent MRI imaging studies that showed pathological lesions compatible with DMG. This assessment was followed by histopathological diagnostic and nuclear immunostaining with recognition of the mutated H3F3A K27M protein. Segmentation of the tumour regions was performed. Images were uploaded into the software LIFEx for postprocessing and extraction of radiomic features.

**Figure 4 diagnostics-13-00849-f004:**
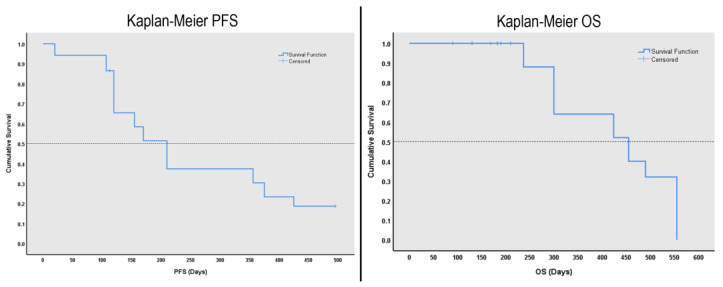
Survival analyses “Kaplan Meier” curves. Part (**left**) represents the curve for PFS. Part (**right**) represents the curve for OS.

**Figure 5 diagnostics-13-00849-f005:**
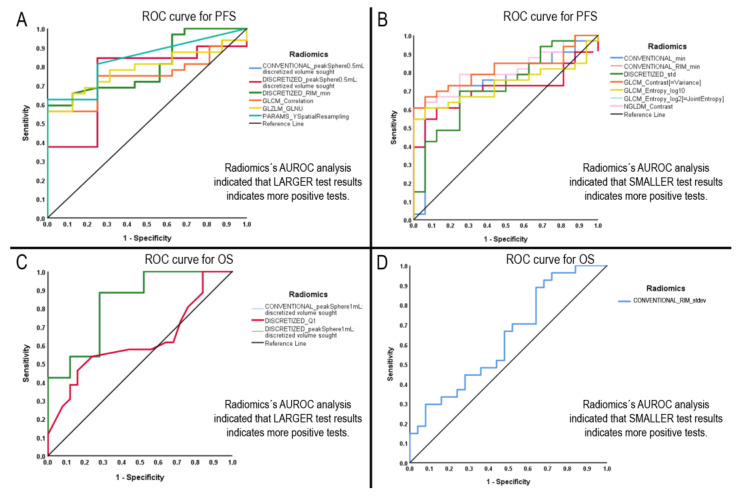
ROC curves of the significant radiomics. Sections (**A**,**B**) show the radiomics with significance for progression-free survival; (**C**,**D**) show the radiomics with significance for overall survival above the median.

**Table 1 diagnostics-13-00849-t001:** Quartiles and IQR for the selected radiomics.

Radiomic Features	Percentile			
	Median	75	25	IQR
CONVENTIONAL_min	128.294	189.296	4.032	185.264
CONVENTIONAL_peakSphere0.5mL: discretised volume sought	0.506	0.522	0.472	0.05
CONVENTIONAL_RIM_min	257.726	337.008	160.108	176.972
CONVENTIONAL_RIM_stdev	70.049	106.502	32.110	74.392
CONVENTIONAL_RIM_Volume(vx)	1528.486	2413.208	783.969	1629.293
DISCRETIZED_std	4.516	5.733	2.995	2.738
DISCRETIZED_Q1	15.500	19.75	10	9.75
DISCRETIZED_peakSphere0.5mL: discretised volume sought	0.506	0.522	0.472	0.05
DISCRETIZED_peakSphere1mL: discretised volume sought	1.041	1.045	1.014	0.031
DISCRETIZED_RIM_min	13.551	250.895	7.077	243.818
PARAMS_YSpatialResampling	0.654	4.864	0.5	4.364
GLCM_Contrast[=Variance]	10.566	22.434	0.046	22.388
GLCM_Correlation	0.606	5.745	0.484	5.261
GLCM_Entropy_log10	2.071	2.368	0.567	1.801
GLCM_Entropy_log2[=JointEntropy]	6.878	7.868	2.230	5.638
NGLDM_Contrast	0.027	0.084	0.001	0.083
GLZLM_GLNU	169.238	68,327.15	65.305	68,261.845

**Table 2 diagnostics-13-00849-t002:** The cut-off value for the significant radiomic features.

Radiomic Feature	Cut-Off Value	AUROC	SE	*p*-Value	95% CI	
					Lower	Upper
Progression-free survival						
*Descriptive characteristics*						
PARAMS_YSpatialResampling	>0.65	0.188	0.064	0.001	0.062	0.313
*First-order characteristics*						
CONVENTIONAL_min	≤100.8	0.763	0.078	0.004	0.611	0.916
CONVENTIONAL_peakSphere0.5mLdiscretizedvolumesought	>0.4	0.232	0.079	0.003	0.078	0.386
CONVENTIONAL_RIM_min	≤244.4	0.683	0.081	0.045	0.523	0.843
DISCRETIZED_peakSphere0.5mLdiscretizedvolumesought	>0.48	0.232	0.079	0.003	0.078	0.386
DISCRETIZED_RIM_min	>14.91	0.683	0.081	0.045	0.523	0.843
DISCRETIZED_std	≤4.85	0.688	0.082	0.040	0.526	0.849
*Second-order characteristics*						
GLCM_ContrastVariance	≤6.12	0.781	0.069	0.002	0.647	0.916
GLCM_Correlation	>0.69	0.290	0.077	0.022	0.138	0.442
GLCM_Entropy_log10	≤1.57	0.685	0.079	0.043	0.530	0.841
GLCM_Entropy_log2JointEntropy	≤5.23	0.685	0.079	0.043	0.530	0.841
GLZLM_GLNU	>232.13	0.246	0.072	0.005	0.104	0.387
NGLDM_Contrast	≤0.02	0.775	0.069	0.003	0.639	0.910
Overall survival						
*First-order characteristics*						
CONVENTIONAL_ peakSphere1mLdiscretizedvolumesought	>1.02	0.134	0.054	<0.001	0.029	0.240
CONVENTIONAL_RIM_stdev	≤106.56	0.686	0.080	0.035	0.529	0.843
DISCRETIZED_Q1	>17	0.321	0.083	0.042	0.160	0.483
DISCRETIZED_ peakSphere1mLdiscretizedvolumesought	>1.02	0.134	0.054	<0.001	0.029	0.240

**Table 3 diagnostics-13-00849-t003:** Diagnostic performance test evaluation for the selected radiomics.

	Radiomic (Significant AUROC)	Sensitivity		Specificity		+LR		−LR		+PV		−PV	
		Value	95% CI	Value	95% CI	Value	95% CI	Value	95% CI	Value	95% CI	Value	95% CI
PFS	CONVENTIONAL_min	63.8%	46.2–79.1%	93.7%	69.7–99.8%	10.2%	1.5–69.2%	0.3%	0.2–0.6%	95.8%	77.2–99.3%	53.7%	42.3–64.4%
	CONVENTIONAL_peakSphere0.5mLdiscretizedvolumesought	97.2%	85.4–99.9%	0%	0–20.5%	0.9%	0.9–1%	-	-	68.6%	67.4–69.8%	0%	-
	CONVENTIONAL_RIM_min	58.3%	40.7–74.4%	93.7%	69.7–99.8%	9.3%	1.3–63.5%	0.4%	0.2–0.6%	95.4%	75.5–99.3%	50%	39.9–60%
	DISCRETIZED_peakSphere0.5mLdiscretizedvolumesought	83.3%	67.1–93.6%	50%	24.6–75.3%	1.6%	1–2.7%	0.3%	0.1–0.8%	78.9%	69.2–86.2%	57.1%	35.6–76.2%
	DISCRETIZED_ RIM_min	63.8%	46.2–79.1%	93.7%	69.7–99.8%	10.2%	1.5–69.2%	0.3%	0.2–0.6%	95.8%	77.2–99.3%	53.5%	42.3–64.4%
	DISCRETIZED_std	63.8%	46.2–79.1%	75%	47.6–92.7%	2.5%	1–6.1%	0.4%	0.2–0.8%	85.1%	70.3–93.2%	48%	35.4–60.7%
	GLCM_ ContrastVariance	52.7%	35.4–69.5%	100%	79.4–100%	-	-	0.4%	0.3–0.6%	100%	-	48.4%	39.9–57.0%
	GLCM_Correlation	52.7%	35.4–69.5%	93.7%	69.7–99.8%	8.4%	1.2–57.7%	0.5%	0.3–0.7%	95%	73.5–99.2%	46.8%	37.9–56%
	GLCM_Entropy_log10	47.2%	30.4–64.5%	100%	79.4–100%	-	-	0.5%	0.3–0.7%	100%	-	45.7%	38.2–53.4%
	GLCM_Entropy_ log2JointEntropy	47.2%	30.4–64.5%	100%	79.4–100%	-	-	0.5%	0.3–0.7%	100%	-	45.7%	38.2–53.4%
	GLZLM_GLNU	52.7%	35.4–69.5%	93.7%	69.7–99.8%	8.4%	1.2–57.7%	0.5%	0.3–0.7%	95%	73.5–99.2%	46.8%	37.9–56%
	NGLDM_Contrast	55.5%	38–72%	93.7%	69.7–99.8%	8.8%	1.3–60.6%	0.4%	0.3–0.6%	95.2%	74.5–99.2%	48.3%	38.9–57.9%
	PARAMS_ YSpatialResampling	58.3%	40.7–74.4%	75%	47.6–92.7%	2.3%	0.9–5.6%	0.5%	0.3–0.8%	84%	68.2–92.7%	44%	33.1–56.3%
OS	CONVENTIONAL_ peakSphere1mLdiscretizedvolumesought	85.1%	66.6–95.8%	60%	38.6–78.8%	2.1%	1.2–3.5%	0.2%	0.09–0.6%	69.6%	58.1–79.2%	78.9%	58.9–90.7%
	CONVENTIONAL_RIM_stdev	85.1%	66.2–95.8%	36%	17.9–57.4%	1.3%	0.9–1.8%	0.4%	0.1–1.1%	58.9%	50.7–66.7%	69.2%	44.1–86.4%
	DISCRETIZED_Q1	51.8%	31.9–71.3%	76%	54.8–90.6%	2.1%	0.9–4.7%	0.6%	0.4–0.9%	70%	51.5–83.6%	59.3%	48.2–69.6%
	DISCRETIZED_ peakSphere1mLdiscretizedvolumesought	85.1%	66.2–95.8%	60%	38.6–78.8%	2.1%	1.2–3.5%	0.2%	0.09–0.6%	69.6%	58.1–79.2%	78.9%	58.9–84.4%

## Data Availability

The data used to support this study’s findings are available from the corresponding author upon reasonable request.

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
