# Peer review of "Diagnostic Performance of Selected MRI-Derived Radiomics Able to Discriminate Progression-Free and Overall Survival in Patients with Midline Glioma and the H3F3AK27M Mutation"

_diagnostics, 2023, doi:10.3390/diagnostics13050849_

Round 1

Reviewer 1 Report

Chilaca-Rosas et al. analyze radiomic signatures that can discriminate PFS and OS in diffuse midline gliomas with H3F3A K27M mutation. Although the paper is interesting, it suffers from a number of limitations listed below, which should be addressed or at least discussed.

1. Can this same signature be applied to a separate cohort using images obtained from a different MRI scanner? I am skeptical as to the robustness of this method.

2.  How about H3F3A/HIST1H3B wildtype gliomas? Are there cases in your cohort with negative H3K27M IHC staining that you can analyze? Downstream H3K27me3 inhibition should be observed in non-H3K27M tumors as well. Are there differences in radiomic signatures?

3. In the abstract, the "O" in overall survival is both capitalized (line 28) and lower case (line 32). Also, numbers less than 10 are both spelled out (line 29- nine) and shown as arabic numerals (lines 21, 29-30). Please be consistent.

4. The authors do not reference the following important paper which looks at MRI radiomic features to predict PFS in a large cohort of 89 pediatric diffuse midline gliomas. This paper should be thoroughly discussed, including similarities and differences between the methods and results.

Wagner et al., Radiomic features based on MRI predict progression-free survival in pediatric diffuse midline glioma/diffuse intrinsic pontine glioma. Can Assoc Rad J 74(1):119-126, 2023.

Reviewer 2 Report

I have the following comments:

-Abstract, lines 29-30. Please avoid generic terms such as "excellent" specificity and "good" sensitivity. Provide quantitative values with full details instead.

Moreover, I suggest deleting the sentence at lines 31-32 ("A statistical trend was found for the role of first-order radiomic features in influencing progression-free survival and overall survival in this orphan entity"), as it may be misleading and in case might find its place in the Results, not in the Conclusions.

-In the Introduction, please expand (1-2 additional sentences) on the main applications of radiomics for cancer care (including its potential use for personalized therapy and prediction of individual treatment outcome), along with its current limitations, e.g., validation issues. Some additional references should be provided to support this part, including doi 10.3389/fpsyg.2021.710982 and doi 10.3390/cancers13092135 among others.

-Materials and Methods, lines 89-91. Please add a table illustrating the details of the MRI acquisition parameters.

Round 2

Reviewer 1 Report

The authors do not make a concerted effort to respond to suggestions made by the reviewer, thus cannot be accepted in the present form. Please answer the following questions.

1. How many patients in the cohort of 91 patients received biopsies?

2. Of the patients receiving biopsies, IHC analysis of Histone H3K27M was performed in how many patients?

3. Of the patients whose tumors were analyzed for H3K27M IHC, how many were positive (was it 12?) and how many were negative?

4. It the answer to the latter part of question 3 (the number of H3K27M-negative patients) is not zero, then analyze these patients MRIs using the same methods as for H3K27M-mutant patients and compare and contrast the results.

Round 3

Reviewer 1 Report

The authors have responded to my questions. I believe this paper is now suitable for publication in Diagnostics.